# Latest Developments in Minimally Invasive Spinal Treatment in Slovakia and Its Comparison with an Open Approach for the Treatment of Lumbar Degenerative Diseases

**DOI:** 10.3390/jcm12144755

**Published:** 2023-07-18

**Authors:** Marina Potašová, Peter Filipp, Róbert Rusnák, Eva Moraučíková, Katarína Repová, Peter Kutiš

**Affiliations:** 1Department of Physiotherapy, Faculty of Health, Catholic University in Ruzomberok, 034 01 Ruzomberok, Slovakia; marina.potasova@ku.sk (M.P.); peter.filipp95@gmail.com (P.F.); rusnakr@uvn.sk (R.R.); repova.katka@gmail.com (K.R.); peter.kutis@ku.sk (P.K.); 2Neurosurgery Clinic, Central Military Hospital SNP in Ruzomberok, 034 01 Ruzomberok, Slovakia; 3Department of Public Health, St. Elizabeth University of Health and Social Work in Bratislava, 811 02 Bratislava, Slovakia; 4Department of Nursing, Faculty of Health, Catholic University in Ruzomberok, 034 01 Ruzomberok, Slovakia; 5Department of Health Care Sciences, Faculty of Humanities, Tomas Bata University in Zlin, 761 01 Zlin, Czech Republic

**Keywords:** spine operative treatment, minimally invasive surgery, LDD, spinal fusion, neurorehabilitation

## Abstract

The study describes the benefits of MIS-TLIF (minimally invasive transforaminal lumbar interbody fusion) and compares them with OTLIF (open transforaminal lumbar interbody fusion). It compares blood loss, length of hospitalization stays (LOS), operation time, and return of the patient to the environment. A total of 250 adults (109 males and 141 females), mean age 59.5 ± 12.6, who underwent MIS-TLIF in the Neurosurgery Clinic (NSC) Ruzomberok, Slovakia, because of lumbar degenerative diseases (LDD), participated in this retrospective study. Data were obtained from the patients’ medical records and from the standardized Oswestry Disability Index (ODI) index questionnaire. To compare ODI in our study sample, we used the Student’s Paired Sample Test. To compare the MIS-TLIF and OTLIF approaches, a meta-analysis was conducted. Confidence intervals were 95% CI. The test of homogeneity (Chi-square (Q)) and the degree of heterogeneity (I^2^ test) among the included studies were used. Statistical analyses were two-sided (α = 0.05). All monitored parameters were significantly better in MIS-TLIF group: blood loss (*p* < 0.001), operation time (*p* < 0.001), and ODI changes (*p* < 0.001). LOS (*p* < 0.042) were close to the significance level. ODI in the study sample decreased by 33.44% points after MIS-TLIF, and it significantly increased as well (*p* < 0.001). The percentage of patients who were satisfied with the surgery they underwent was 84.8%. The study confirmed that the MIS-TLIF method is in general gentler for the patient and allows the faster regeneration of patient’s health status compared to OTLIF.

## 1. Introduction

Transforaminal lumbar interbody fusion (TLIF) has been widely performed for lumbar degenerative disease (LDD) [1]. TLIF was first introduced to overcome the potential risk of nerve root injuries and dural tears associated with posterior lumbar interbody fusion [2]. Today, it is the most frequently used method in spinal surgery. TLIF can be performed in two ways, as open or mini-invasive surgery [3].

A long-used surgical technique has been open transforaminal lumbar intervertebral fusion (OTLIF). In OTLIF, one larger incision is made in the center of the spine, and a unilateral transforaminal approach is used to insert the intervertebral disc replacement, although many surgeons may prefer bilateral decompression. In addition, bilateral transpedicular fixation is performed [4].

Currently, minimally invasive surgical techniques, such as minimally invasive transforaminal lumbar intervertebral fusion (MIS-TLIF), are gaining popularity in spinal surgery. 

During minimally invasive transforaminal lumbar interbody fusion (MIS-TLIF), the outcome of the decompression and fusion appear to be similar to the open approach (Khashab 2023 [3]). However, MIS-TLIF has evolved to decrease muscle dissection and operative morbidities associated with open surgery [5]. Thus, MIS-TLIF allows less tissue injury to the posterior paraspinal muscle groups, especially multifidus, and preserves posterior spinal midline structures. It also enables other benefits such as smaller incisions, lesser blood loss, shorter hospital stays, faster post-operative recovery, earlier return to work, and hence better functional outcomes [6,7,8,9]. 

### 1.1. MIS-TLIF in Diagnostics and Treatment

Patient diagnosis consists of an objective neurological examination of the patient and an anamnesis of the patient’s subjective complaints. The next step consists of imaging methods. We use dynamic X-ray examination in forward and backward bending and magnetic resonance. Dynamic X-ray examination evaluates the instability or spondylolisthesis of the lumbar spine. Magnetic resonance evaluates degenerative disc disease, spinal stenosis, or foraminal stenosis. According to complex examinations, surgeons can decide on instrumented spine surgery.

In MIS-TLIF, decompression and intervertebral replacement are performed through tubular retractors or expanders, followed by percutaneous transpedicular fixation. With this technique, two smaller incisions are made on each side of the lumbar spine to provide access to the spine with minimal disruption of muscle tissue [10,11,12]. 

In our clinic, all MIS-TLIF surgical techniques are performed under general anesthesia. During the procedure, the patients are in the prone position on an operative table. We use standard 2D fluoroscopy for intraoperative imaging and a surgical microscope for visualization purposes. The locations of the incisions used for inserting the retractor are paraspinal/paramedial. We use both tubular and nontubular expandable retractors. Then, we perform the decompression and interbody cage insertion first, prior to the pedicle screw and rod placement. This decompression includes complete facetectomy, nerve root decompression, discectomy, and interbody cage insertion with laminectomy prior to pedicle screw insertion. The facetectomy is always performed on the symptomatic side, and we pack the interbody cage with graft material [13].

### 1.2. The Aim of the Study

The aim of the study was to verify the benefits of MIS-TLIF such as the minimization of blood loss, operation time, and length of hospitalization stays (LOS), as well as to verify the patients’ return to their family, work, and social environment, by measuring the Oswestry Disability Index (ODI). Last, but not least, the aim of the study was to compare the benefits of MIS-TLIF with an open approach (OTLIF). At The NSC Ruzomberok, only MIS-TLIF has been preferred for the last 10 years. OTLIF is no longer implemented (with a few exceptions that are not statistically significant). Therefore, we used meta-analysis and the available literature to compare our results with the OTLIF technique.

## 2. Materials and Methods

A retrospective study was undertaken.

### 2.1. Study Participans

A total of 250 adults (109 males and 141 females; age: 59.5 ± 12.6 years) voluntarily participated in the current research. These were patients from the Ruzomberok Neurosurgical Clinic who underwent MIS-TLIF due to lumbar degenerative disease (LDD) during the period of January 2022–November 2022. The indication for MIS-TLIF, as well as the procedure itself, was performed by 5 experienced neurosurgeons working at our clinic. The diagnosis of LDD, the indication for surgery, and the MIS-TLIF technique were performed according to the procedures described in the introduction. The inclusion criteria and exclusion criteria for study participants are presented in Table 1.

The exclusion criteria in points (a) and (b) were established due to the patients’ inability to identify the cause of the pain and its direct relation to MIS-TLIF surgery.

All subjects received detailed information about the objectives, benefits, and risks associated with participation in this study. They also signed an informed consent form indicating their willingness to participate in the current research.

### 2.2. Data Collection

The selection of potential patients was established through post-operative database. The data collection process had three phases. The first phase was focused on collecting basic data from patients’ medical records. Age, gender, surgery technique, operated segment, blood loss, operative time, and LOS data were collected retrospectively. Simultaneously with the first phase, the second phase took place, in which we evaluated the pre-operative Oswestry Disability Index (ODI). Patients filled the pre-operative ODI face-to-face with the medical staff 1 day before surgery. ODI was part of the patients’ medical documentation. In the third phase, the patients were contacted by phone and answered questions from the standardized ODI. In the third phase, we obtained information about patients’ post-operative health conditions.

A total of 284 patients were approached. Of these 284 patients, 34 patients were excluded because they did not meet the inclusion criteria. Reasons for exclusion: *n* = 14 hip and knee alloplastic at the time of post-operative data collection; *n* = 2 falls and subsequent worsening of pain at the time of post-operative data collection; *n* = 1 death; *n* = 1 internal medicine problems and hospitalization at the time of post-operative data collection; *n* = 6 did not agree to participate in the study; and *n* = 10 did not answer the phone. Therefore, the final sample was composed of 250 patients (Figure 1).

### 2.3. OTLIF Sample Selection

We reviewed the online database PubMed using the keywords ‘lumbar,’ ‘degenerative diseases,’ ‘open,’ and ‘minimally invasive ‘. We first screened all articles using their abstracts, and we included only English-language reports with full text manuscripts. Additional inclusion criteria included: (1) studies with populations consisting of adult patients > 18 years of age, (2) studies including a group of patients treated with OTLIF, (3) studies comparing at least one desirable outcome (e.g., operative time, blood loss, LOS, ODI), (4) studies where calculated the arithmetic mean, sample standard deviation and sample range from the data for each patient. After applying our inclusion criteria, 25 articles were chosen for comparison.

### 2.4. Ethical Clearance

This study was conducted in the accordance with principles outlined in the Helsinki Declaration. It was also approved by The Ethic Committee in Central Military Hospital in Ruzomberok, Slovakia (approval no. ÚVN-56-20/2023).

### 2.5. Method

In this study, we have chosen the method of retrospective data collection from the patients’ medical record and evaluation of the standardized ODI index. Blood loss was analyzed in milliliters (mL), LOS in days, and operation time in minutes (min). In the ODI, patients answered 10 questions, which we evaluated on a scale of 0 to 5 points (Appendix A). Based on the total score, we evaluated the patients’ pre-operative and post-operative health status as follows: 0–20% No disability; 21–40% moderate disability; 40–60% severe disability; 61–80%—crippled; 80–100% complete disability [14]. At the end of the ODI questionnaire, patients answered 3 questions: Did the operation meet their expectations? Have they completed post-operative rehabilitation? Have they completed a spa treatment? With these questions, we monitored patients’ subjective satisfaction with the operation, and we followed whether they used comprehensive post-operative care to support the effectiveness of the operation.

Every patient who underwent MIS-TLIF surgery underwent post-operative rehabilitation as well. Rehabilitation consists of kinesiotherapy and ergotherapy. Kinesiotherapy includes exercises for maintaining muscle strength and mobility of the upper and lower limbs, isometric strengthening of trunk muscles, respiratory and vascular gymnastics, verticalization, and mobilization. All exercises are performed with no movements in the lumbar spine. This ensures the regeneration and prevents damage to the operated section. Ergotherapy is also indicated as a part of the post-operative rehabilitation. The patient is trained to perform basic daily activities, such as putting on their shoes, dressing the lower part of the body, lifting loads, getting up and getting into the bed, personal hygiene, etc. Daily activities are performed with an emphasis on not overloading the operated section. The patient is rehabilitated once a day, lasting 20–30 min, during hospitalization.

Six weeks after the first post-operative control, a neurosurgeon is debilitated for additional rehabilitation. Patients can choose an ambulant or institutional form of rehabilitation. In later forms of rehabilitation, the patient can undergo kinesiotherapy, myofascial techniques, scar treatment, hydrotherapy, etc. In addition, the patient is also provided with spa treatment, which is covered by public health insurance. 

### 2.6. Statistical Analysis

We analyzed the data obtained in our study sample with descriptive statistics. In the results, we present the frequency, percentage, mean, and standard deviation. The paired Student’s test was used to compare pre-operative and post-operative ODI indexes (*p* < 0.05).

To compare MIS-TLIF and OTLIF approaches, the meta-analysis was launched in IBM SPSS 28. A formal analysis was conducted for all outcomes if the dates were sufficient. The input for the analysis was the data from our sample set, where we calculated the arithmetic mean, sample standard deviation, and sample range from the data for each patient. This group was considered as treatment group for all comparisons. We compared these parameters with samples from the chosen different studies, from which we selected data for the control group (OTLIF), e.g., mean, standard deviation, and number of patients (respectively sample variance). Confidence intervals were 95% confidence interval (CI), while pooled continuous effect measures were expressed as the mean difference with 95% CI. When interpreting the results of a meta-analysis, it is important to consider both the statistical significance and the magnitude of the effect size estimate (test of homogeneity—Chi-square (Q statistic)), as well as the degree of heterogeneity among the included studies. We explored and quantified between-study statistical heterogeneity using the I^2^ test. By default, we used the fixed-effect model in all analyses. If heterogeneity was statistically significant (*p* < 0.05) or I^2^ was >50%, heterogeneity is considerable across the studies, and the results should be considered cautiously. Statistical analyses were two-sided, with an α-error of 0.05. Results are presented in tables and in graphs.

#### Description and Interpretation of Tables

The results in the tables show the following:The label, which represents the name of the study: the first author + year.Effect size, which was calculated as the mean difference between two groups + the 95% confidence interval.Standard error, which provides a measure of the precision of the effect size estimate. A smaller standard error, which indicates that the sample estimate is more precise and is more likely to be closer to the true population value.The *p*-value (Sig.), which provides the significance of the comparison of the control (OTLIF) and the treatment (our MIS-TLIF) group.The weight of each study, as a percentage of the total of the meta-analysis (100%).The overall results, which are represented by Z and *p* values. When *p* is < 0.05, the overall result is statistically significant.The last line written in the table indicates the heterogeneity represented by the I^2^ values and homogeneity represented by Q (Chi-square) and *p* value.The overall results are represented by Z and *p* values. When *p* is < 0.05, the overall result is statistically significant.The last line written in the table indicates the heterogeneity represented by the I^2^ values and homogeneity represented by Q (Chi-square) and *p*-value.

## 3. Results

In the following section, we present the analysis of the results. In the first part, we analyze the results of our study sample. In the second section, we compare the results obtained from our study sample with the control group (patients from chosen studies). These sections should provide a concise and precise description of the experimental results, their interpretation, and the experimental conclusions that can be drawn.

### 3.1. Analysis of MIS-TLIF Results in the Study Sample

We analyzed the results obtained from our study sample. Results are presented in Table 2.

A total 109 men and 141 women participated in the study, and 48.8% of the monitored patients underwent multi-level MIS-TLIF. Single-level MIS-TLIF was performed in 51.2% of the operated patients. The most commonly treated levels were L4/5/S1. Multiple MIS-TLIF (L4/5/S1) was implemented in 91 patients (36.4%). In the monitored group, the average blood loss was 265.73 ± 104.39 mL, the average LOS was 4.84 ± 0.81 days, and the average operative time was 107.69 ± 29.12 min. The post-operative ODI index was adjusted by 33.44% points, and at the same time, there was a significant improvement in the post-operative ODI index, which reflects a significant improvement in the overall health status of patients after completing MIS-TLIF (*p* < 0.001). Satisfaction with undergoing surgery was reported by 212 (84.8%) of patients. Post-operative ambulance or institutional rehabilitation was completed by 179 (71.6%), and spa treatment was completed by 78 (31.2%) of patients.

### 3.2. Blood Loss Analyses

Table 3 presents the results of 17 different studies dealing with blood loss in OTLIF. We compared these results with the results obtained in our study sample. The results are arranged according to the effect size in descending order. We present the graphical results in Figure 2.

The calculated data present the Effect Size Estimates for Individual Studies (blood loss). The data show that the blood loss (mL) of patients after the MIS-TLIF is significantly lower than blood loss within OTLIF. This was confirmed by comparing our treatment group with control samples in 15 different studies [15,16,17,18,19,20,21,22,23,24,25,26,27,28,29] was not significant.

Effect Size Estimates provide us with the overall results. The overall mean effect size is −331.234 mL. This value represents the point estimator of the average reduction in blood loss (mL) with the MIS-TLIF method. Its 95% CI from −432.013 to −230.454 gives us the interval estimation for reduction in the blood loss. The *p*-value is < 0.001. Therefore, the overall result is statistically significant. The obtained results confirm that blood loss during MIS-TLIF is significantly lower than during OTLIF.

These results also confirm the heterogeneity of the given studies (Appendix A).

### 3.3. LOS Analyses

Table 4, Effect Size Estimates for Individual Studies (LOSs), shows the comparison of the results between control group and our MIS-TLIF group. 

The calculated data show that LOS (days) was significantly shorter in MIS-TLIF group compared to the OTLIF control group in only nine studies [10,11,12,13,14,15,15,16,17,17,18,19,20,22,24,25,26,31]. In two studies, we have comparable results [21,28]. In the last three studies, we can see significantly shorter LOS in the control group compared to our treatment group (Figure 3). So there is a significant difference in the opposite way.

The point estimate for the overall mean effect size is −2.776 days when the patient is operated on with the MIS-TLIF approach. Its 95% CI is from −5.449 to −0.103 days of hospitalization. The *p*-value < 0.042 is close to the significance level of 0.05. However, the overall result is statistically significant. The obtained results closely confirm that the length of hospitalization after the MIS-TLIF method is used is significantly lower than after the open OTLIF method is used.

Of course, we again have a high degree of heterogeneity (Appendix A).

### 3.4. Operative Time Analyse

Table 5, Effect Size Estimates for Individual Studies (Operative time (min)), shows the comparison of the results between control group and our treatment group. 

The calculated data show that operative time (min) was significantly shorter in our MIS-TLIF group compared to all the control groups with OTLIF (Figure 4). The differences in means (Effect Size) were from −171.120 min to −28.320 min. All differences are significant with a *p*-value < 0.001.

The overall mean effect size was −70.269 min. This value represents the point estimator of the average reduction in operative time (min) with the MIS-TLIF method. Its 95% CI from −87.609 to −52.929 gives us the interval estimation for the reduction in operative time in minutes. The *p*-value is < 0.001. So the overall result is statistically significant. The obtained results confirm that operative time during MIS-TLIF is significantly shorter than during OTLIF.

We again have a higher degree of heterogeneity (Appendix A).

### 3.5. ODI Change Analyses

As the last parameter, we compared changes in ODI. We decided to analyze the ODI changes, as the input and output ODI indexes in groups were incomparable (in our file, ODI input data were significantly higher than in control groups). With the decision to compare the change in ODI, we had to challenge the question of how to obtain the right values of the average mean and standard deviation for the change parameter. The studies used provided us with only the mean and standard deviation for the ODI Perioperative and ODI Follow-up. We calculated the average mean for the change as a difference between the given means. The calculation of standard deviation is not so straightforward. We used the Monte Carlo method for the estimation of this parameter. However, this method is only an approximation for missing data. Regarding the dependence of ODI values (input and output), we suppose that the calculated standard deviations represent an upper estimation of the real standard deviation for ODI change. The data are presented in Table 6.

The calculated data are presented in Figure 5. Figure 5 shows that the change in ODI is higher in our treatment group compared to 9 control groups [15,17,20,22,33,34,35,36,37]. Compared to four studies, our treatment group has comparable improvement in ODI [23,26,32,38]. The value of the Effect Size represents the difference between the Change in ODI mean (ODI pre-operative to ODI Post-operative) in the control group and our treatment group. The OTLIF method compared to these studies represents, in general, a decrease in ODI measurement from 4.828 to 24.138 points. And these differences are significant, with *p*-values from 0.000 to 0.004. Compared to the last fur studies, our treatment group showed a comparable improvement in ODI like control groups. Their *p*-values are from 0.079 to 0.691.

The overall results present the average ODI Change in the MIS-TLIF method as 9399 points better than in OTLIF. Its 95% CI is from 5.677 to 13.121. The *p*-value is < 0.001. So the overall result is statistically significant. The obtained results confirm that the change in ODI after the MIS-TLIF method is significantly better than after he OTLIF method.

We meet again with a higher degree of heterogeneity (Appendix A).

## 4. Discussion

This study investigates the benefits of minimally invasive spine surgery and compares them with the OTLIF approaches.

The most mentioned benefits of minimally invasive surgery are reduced trauma of the tissue, lower blood loss, shorter operative time and LOS, reduced intraoperative radiation exposure, earlier verticalization and mobilization of patients and their earlier integration into the daily activities or the environment, lower operating costs, fewer post-operative medications or complications, and others [39,40]. There are many benefits of minimally invasive spine surgery. So in the present study, we analyzed those, which are the most often recorded in our clinic. We analyzed four benefits: blood loss, LOS, operative time, and patients’ post-operative health status by measuring ODI index.

Due to the fact that OTLIF approaches have not been implemented at NSC Ruzomberok for almost 10 years (only in exceptional cases, which are statistically insignificant), we present the obtained results at two levels: in the first level, we analyzed the results obtained in our study sample, and in the second, we compared them with the selected literature dealing with OTLIF approaches using meta-analysis tools.

### 4.1. Main Findings Obtain in Study Sample

The results obtained in our study sample confirm that MIS-TLIF approaches significantly improve the overall health status of the patients. This fact was confirmed using the improved mean post-operative ODI index by 33.44% points, as well as using the analysis of the pre- and post-operative ODI using the Student’s *t*-test. The improvement in post-operative ODI was significant (*p* < 0.001). In addition to the MIS-TLIF technique itself, ODI improvement was also a result of post-operative rehabilitation. Tarnanen [41] analyzed disability using ODI and muscle strength in patients undergoing lumbar spine fusion. They analyzed associations between changes in trunk muscle strength and disability after spinal fusion and post-operative rehabilitation. The pre-operative extension/flexion strength ratio was 0.79 in females and 0.76 in males. Three months post-operatively, the strength ratio decreased to 0.66 in males (*p* = 0.02). The mean ODI improved by 47%, and back pain decreased by 65% (both *p* < 0.001). The changes in the ODI correlated with changes in trunk extension (r = −0.38) and flexion (r = −0.43) strength [41]. So changes in ODI were related to spinal fusion and post-operative rehabilitation [41].The same results were confirmed in our study. Oestergaard et al. 2012 examined the effect of early initiation of rehabilitation after instrumented lumbar spinal fusion by measuring ODI, too. According to the ODI, at 1-year follow-up, the 6-week-group had a median reduction of −6 (−19; 4) compared with −20 (−30; −7) in the 12-week group (*p* = 0.05) [18]. Rehabilitation techniques after spinal fusion surgery showed reduced inflammation, decreased pain, improved blood circulation, reduced swelling, lengthened short or tight connective tissue, relaxed tense muscles, soothed nervous system, and facilitated patient recovery [42]. Patient education, which is part of post-operative rehabilitation, reduces anxiety and increases patient satisfaction [43]. In our study, all 250 patients were educated about the post-operative regimen. A total of 212 patients reported satisfaction with the MIS-TLIF and post-operative care, which represents 84.8%. So, the study confirms the improvement in the patient’s health status and satisfaction with ongoing MIS-TLIF surgery. These findings are not only the results of the MIS-TLIF itself but of the early post-operative rehabilitation, too.

In the monitored study sample, the mean blood loss was 265.8 ± 104.8 mL, the mean LOS was 4.84 ± 0.81 days, and the mean operative time 107.8 ± 29.23 min. Further, we compared these results with the control group selected from the literature dealing with OTLIF approaches.

### 4.2. Blood Loss

Blood loss was the first parameter we compared. The obtained results confirm that the blood loss in the OTLIF control group was significantly higher than in the MIS-TLIF treatment group. This was confirmed by comparing results from our treatment group with the control group represented in 17 different studies. Out of these 17 different studies, in 15 studies, blood losses were significantly higher [15,16,17,18,19,20,21,22,23,24,25,26,27,28]. Only the comparison with the control groups of 2 studies were not significant [29,30]. However, in our study, the *p*-value for the overall result was <0.001, so the overall result is statistically significant. In addition to the studies where we compared our results, significantly lower blood losses in MIS-TLIF compared to OTLIF report 8 other studies dealing with this issue [44,45,46,47,48,49,50].

However, when interpreting the results of a meta-analysis, it is important to consider both the statistical significance and the magnitude of the effect size estimate (test of homogeneity), as well as the degree of heterogeneity among the included studies. *p*-value < 0.001 for the overall effect size (Table 3) shows that the decrease in blood loss is unlikely to be caused only by random effects and that there exists another factor that can affect this decrease. The values of I^2^ = 97.07%, Q = 714.992, and *p*-value = 0.000 confirm the heterogeneity of the given studies. So the other factors can have an impact on blood loss, not only on the MIS-TLIF method itself. According to our experience, blood loss is usually higher in central spinal canal stenoses, where we can perioperatively find larger vascular varices that are bloodier. These varices depend on the height of the lumbar segment. So the blood losses can be higher in cranial segments of the lumbar spine. Likewise, higher blood loss can be associated with patients’ comorbidities, such as arterial hypertension or coagulopathies. Last but not least, the factor that contributes the amount of blood loss is the number of segments that are treated. This was confirmed by comparisons of our MIS-TLIF group with other studies dealing with MIS-TLIF approaches. In our MIS-TLIF group, up to 48.8% of patients had multi-level MIS-TLIF, compared to five studies dealing with one level of MIS-TLIF. In single-level operations, the mean blood loss during MIS-TLIF ranged from 100 mL to 208 mL [3,9,20,47,50]. In our MIS-TLIF group, the mean blood loss rate was 265.8 ± 104.8 mL. This value is higher than in the mentioned studies. Only one study, dealing with one-segment MIS-TLIF with higher mean blood losses (456 mL), was reported in the MIS-TLIF group [48]. So we can conclude that the number of treated segments can have an impact on blood loss, not only on the TLIF method itself.

Nevertheless, due to the fact that we find significant differences in blood losses in 15 comparisons out of 17, we consider these results as beneficial and worthy of attention. We conclude that the MIS-TLIF method has a significant effect on blood loss.

### 4.3. LOS

The second analyzed parameter was LOS. The calculated data showed that LOS was significantly shorter in our MIS-TLIF group compared to the OTLIF in only nine studies [10,11,12,13,14,15,15,16,17,17,18,19,20,22,24,25,26,37]. We have comparable results in two studies [26,33]. In three studies [18,27,30], LOS was significantly shorter than in our treatment group. The *p*-value of 0.042 for the overall effect size is significant, but only very narrowly. Of course, we meet the higher value of I^2^ = 99.08% again. Again, this means a high degree of heterogeneity. This was also confirmed using the test of homogeneity, with Q = 4420.336 and *p*-value = 0.000. As mentioned above, these results suggest that the studies included in the meta-analysis may have differed significant in their methods, populations, or other factors that could have impacted the effect size estimate. Considering the results for *p*-value and I^2^ and the other parameters, we suppose that the length of hospitalization is influenced by the system of healthcare in different countries more than other factors. In addition to the studies where we compared LOS in MIS-TLIF and OTLIF, statistically significantly shorter LOS in MIS-TLIF was reported in six other studies, where authors compared the MIS-TLIF and OTLIF [9,20,21,45,46,48,50]. In these studies, the mean LOS for OTLIF was 8 to 12.5 days. In our study, mean LOS for MIS-TLIF was 4.84 ± 0.81 days. Tarman, for example, reported the mean LOS for OTLIF for only 3 days, which is shorter than not only our MIS-TLIF group, but also all the above-mentioned studies. However, when we compare our results with studies dealing with MIS-TLIF, we can see various LOSs for the MIS-TLIF technique as well. We found eight studies where the mean LOS for MIS-TLIF was from 2 to 6.4 days [3,9,20,21,45,46,48,50]. In one study even reported a mean LOS for MIS-TLIF of 9.3 days [48].

Besides the health care systems in different countries, LOS can be affected by post-operative rehabilitation, too. MIS-TLIF enables less intervention in the integrity of the organism, so this allows early mobility (transfer training and supervised gait training) in acute care. So early mobility consequently reduces LOS [19].

Therefore, we emphasize once again the fact that LOS may be affected by the health systems of different countries or early post-operative rehabilitation as well. So we are very cautious in stating that MIS-TLIF has a significant impact on the length of hospitalization.

### 4.4. Operative Time

In the next step, we evaluated the length of the operative time. Operative time (min) was significantly shorter in our MIS-TLIF group compared to all control groups with OTLIF approaches [15,16,24,25,26,27,28,29,30,31,32]. The *p*-value (<0.001) for the effect size was very significant. But again, we have a higher value of I^2^ = 97.6% and a high degree of heterogeneity. This was confirmed by a test of homogeneity with Q = 637.156 and *p*-value = 0.000. The length of the operative time suggests that the studies included in the meta-analysis deal with different factors that could impact the effect size estimate. For example, the operative time can be influenced by patients’ clinical condition and comorbidity or surgeon experience. When we mention the surgeon experience as a factor influencing the operative time, we can appeal to the fact that 7 seven studies published from 2007 to 2019 reported significantly shorter operative time with OTLIF approaches compared to MIS-TLIF (mean operative time for OTLIF, 102–365 min; for MIS-TLIF, 144–390 min) [9,20,32,44,45,46,47,48]. Hu et al. [1] state that this difference could be caused by the MIS-TLIF technical learning curve as a new technique, while OTLIF was mastered fluently. They state that MIS-TLIF in these cases studies the initial learning cases, while OTLIF was familiar to surgeons. So if the surgeon masters the skills and gains adequate experience in MIS-TLIF, the operative time can become almost equal to that in the OTLIF group [1]. Today, researchers report significantly shorter operative time in MIS-TLIF. This confirms our study, too. In our study, the mean operative time for MIS-TLIF was 107.8 ± 29.23 min, which is less than the mean time for OTLIF in the six mentioned studies from previous years. Only Brodano et al., 2015 report an operation time of 144 min in OTLIF, which is shorter than in our MIS-TLIF group [32].

Nevertheless, given the fact that, in our study, the MIS-TLIF group was significantly shorter than in all control OTLIF groups, we consider these results as beneficial and worthy of attention. We conclude that MIS-TLIF has a significant effect on operative time.

### 4.5. ODI Change

Last but not least, we compare patients’ health status by measuring ODI for the MIS-TLIF and the OTLIF groups. Due to the unquotability of input data for ODI in the MIS-TLIF and the OTLIF groups, instead of comparing pre-operative and post-operative ODI, we analyzed ODI changes. ODI changes were higher in our treatment group compared to 9 studies [15,17,20,22,33,34,35,36,37] Compared to four studies, our treatment group has comparable improvement in ODI [23,26,32,38]. Overall, the *p*-value for ODI changes reaches <0.001. So the overall result is statistically significant. Three to twelve months post-operative ODI in MIS-TLIF is significantly lower than in OTLIF. Similar results are reported by 4 different studies [51,52,53,54] who state that, 2 years post-operatively, the ODI scores are significantly lower in the MIS-TLIF group than in the OTLIF group. But no significant differences were found in ODI scores between the two groups at 10 years post-operatively [54]. Based on our experience, the health status, as well as the ODI changes, are better adjusted in patients with a shorter duration of neurological symptoms. Acute and subacute symptoms are better treated than chronic ones. We can explain this based on the duration of nerve damage. With a shorter duration of damage, the nerve structures require a shorter time for regeneration after surgical decompression, and therefore patient’s health status is rather improved. During the chronic phase of LDD, neuropathic pain prevails, and health status is more difficult. It adjusts slower regeneration after an operation, too. So these facts may influence ODI changes as well. The ideal surgical decompression of nerve structures is within two years from the onset of the first neurological symptoms. The compression of nerve structures lasting longer than two years results in an anatomical structure change [55].

Overall, MIS-TLIF approaches have proven to be gentler for the patient compared to OTLIF approaches, enabling a return of the patient to their family, social life, and work environment. We must appeal the fact that the patient’s return to environment is guaranteed not only by the operative method itself, but also by complex post-operative care emphases on neurorehabilitation and occupational therapy.

### 4.6. A Strength of the Study

We consider the description of MIS-TLIF with complex post-operative rehabilitation to be a strength of the study. Previous studies have analyzed only the MIS-TLIF technique itself or have described rehabilitation for spinal fusions. There is a lack of studies describing rehabilitation in MIS-TLIF. However, there is a lack of analyses of the benefits of MIS-TLIF from the point of post-operative rehabilitation, which ultimately affects the surgical performance itself. However, any excellent surgical performance may be underappreciated if this work is not followed by complex post-operative care and rehabilitation. Therefore, the main finding of the study is the fact that the benefits of MIS-TLIF (especially LOS and ODI changes) are the result of the complex post-operative care, not just of the surgical technique itself.

### 4.7. Limitation of the Study

Finally, it is necessary to mention the limitation of the study. We consider the main limitation to be the impossibility of comparing MIS-TLIF and OTLIF approaches directly in our clinic. As we already mentioned, OTLIF is not implemented at the NSC in Ruzomberok, and we therefore had to compare the results obtained in our study sample with the literature dealing with OTLIF research. This resulted in heterogeneity in the file. But despite this, the observed benefits of MIS-TLIF approaches were confirmed as significant in all monitored parameters.

## 5. Conclusions

The study confirmed that the MIS-TLIF method is in general gentler for the patient and allows his faster return to the environment, compared to OTLIF approaches. These findings were confirmed by significantly better results in blood loss, operative time, ODI changes, and LOS in MIS-TLIF group compared to OTLIF. The benefits of MIS-TLIF were also confirmed by improved post-operative ODI index in our study sample (ODI decreased by 33.44% points), as well as with patients’ own satisfaction with this surgical procedure, which was declared by 84.8% of patients who underwent it. At the same time, we must state that the patients’ return to daily activities was supported by complex post-operative rehabilitation and ergotherapy. So, the benefits of MIS-TLIF (especially LOS and ODI changes) are the result of the complex post-operative care, not just of the surgical technique itself.

## Figures and Tables

**Figure 1 jcm-12-04755-f001:**
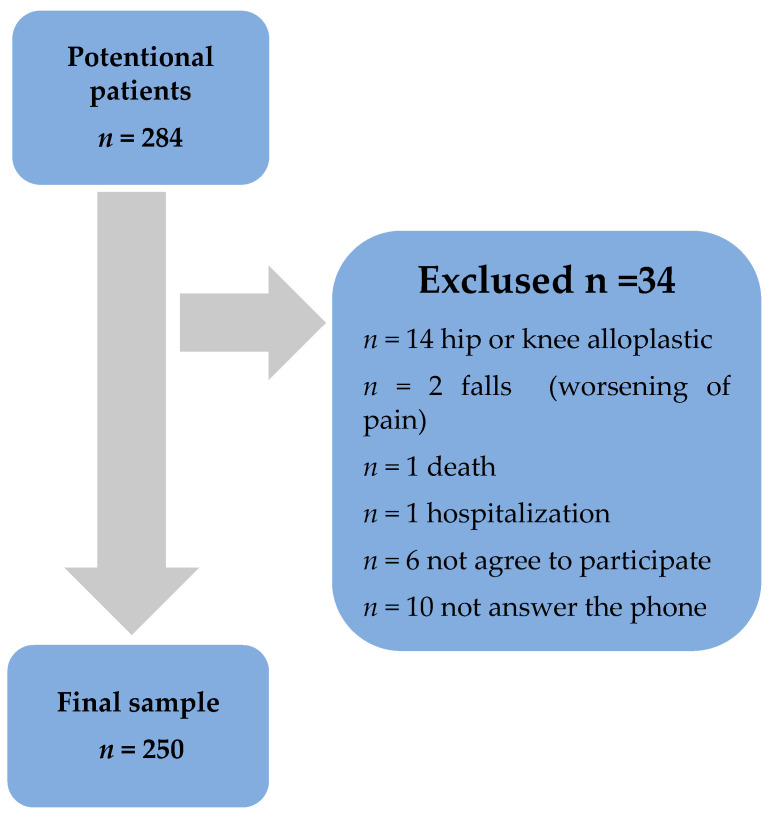
Study sample selection.

**Figure 2 jcm-12-04755-f002:**
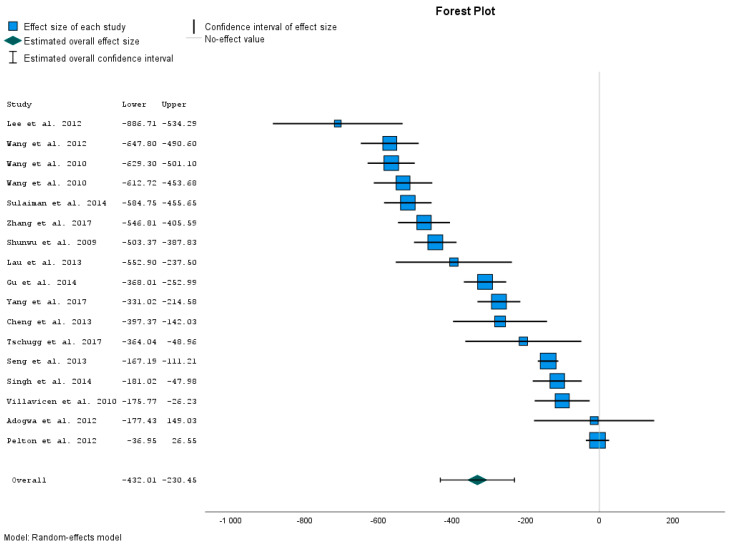
Blood loss comparison [15,16,17,18,19,20,21,22,23,24,25,26,27,28,29,30].

**Figure 3 jcm-12-04755-f003:**
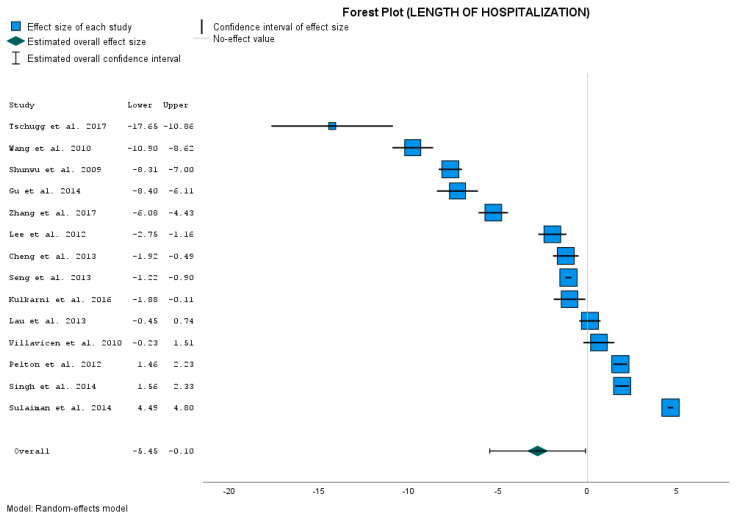
LOS comparison [15,17,18,19,20,21,22,24,25,26,27,28,30,31].

**Figure 4 jcm-12-04755-f004:**
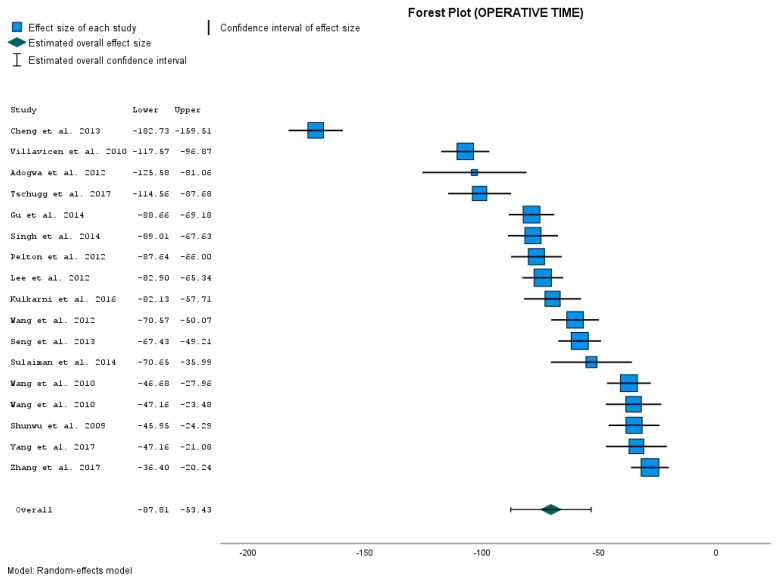
Operative time comparison [15,16,17,18,19,20,22,23,24,25,26,27,28,29,30,31].

**Figure 5 jcm-12-04755-f005:**
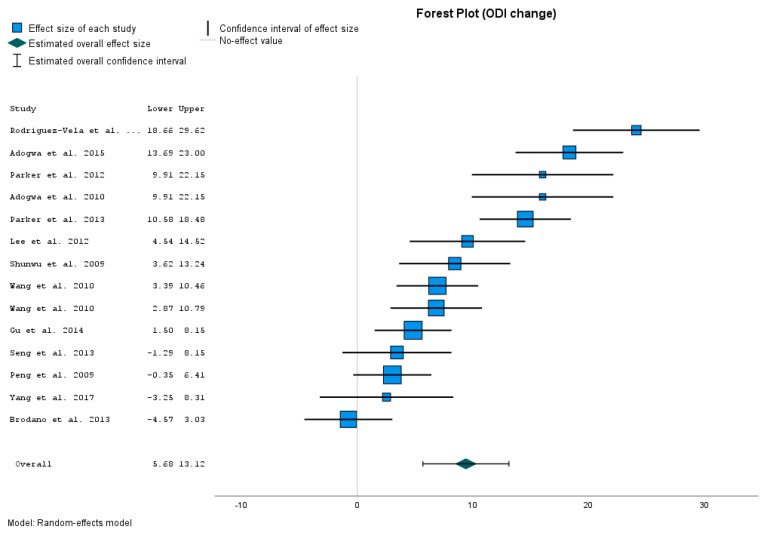
ODI change comparison [15,17,20,22,23,26,32,33,34,35,36,37,38].

**Table 1 jcm-12-04755-t001:** Inclusion and Exclusion Criteria.

Inclusion Criteria			Exclusion Criteria
a.	adult patients	a.	hip or knee alloplastic or injuries at the time of post-operative data collection
b.	3 to 12 months after MIS-TLIF	b.	musculoskeletal pain at the time of post-operative data collection
c.	indication for surgery—LDD	c.	unwillingness to participate in the study
d.	consent to participation in the study		

**Table 2 jcm-12-04755-t002:** Basic Analysis of the Obtained Data (N = 250).

	N	%	Mean	SD	*p*
M/F	109/141	43.6/56.4	-	-	-
MIS-TLIF in L4/5/S1	91	36.4	-	-	-
MIS-TLIF in L3/4/5	27	10.8	-	-	-
MIS-TLIF in L2/3/4	4	1.6	-	-	-
MIS-TLIF in L5/S1	61	24.4	-	-	-
MIS-TLIF in L4/5	53	21.2	-	-	-
MIS-TLIF in L3/4	9	3.6	-	-	-
MIS-TLIF in L2/3	4	1.6	-	-	-
MIS-TLIF in L1/2	1	0.4	-	-	-
Blood loss	-	-	265.8	104.8	-
LOS	-	-	4.84	0.81	-
Operative time	-	-	107.8	29.23	-
P*re- and post- operative* ODI [%] post-operative	-	-	33.44	23.749	<0.001
Satisfaction with MIS-TLIF	212	84.8	-	-	-
Underwent rehabilitation	179	71.6	-	-	-
Underwent spa	78	31.2	-	-	-

**Table 3 jcm-12-04755-t003:** Effect Size Estimates for Individual Studies (Blood Loss [days]).

Study	MIS-TLIF Mean ± SD (*n* = 250)	OTLIF Mean ± SD (*n*)	Effect Size	Mean Diff, 95% CI	Sig. (2-Tailed)	Weight (%)
Lee et al., 2012 [15]	265.8 ± 104.8	976.3 ± 760.8 (*n* = 72)	−710.5.	(−886.71; −534.29)	<0.001	5.2
Wang et al., 2012 [16]	265.8 ± 104.8	835.0 ± 247.0 (*n* = 39)	−569.2	(−647.80; −490.60)	0.000	6
Wang et al., 2010 [17]	265.8 ± 104.8	831.0 ± 210.0 (*n* = 43)	−565.2	(−629.30; −501.10)	0.000	6.1
Wang et al., 2010 [17]	265.8 ± 104.8	799.0 ± 208.0 (*n* = 27)	−533.2	(−612.73; − 453.68)	0.000	6
Sulaiman et al., 2014 [18]	265.8 ± 104.8	786.0 ± 107.0 (*n* = 11)	−520.2	(−584.75; −455.65)	0.000	6.1
Zhang et al., 2017 [19]	265.8 ± 104.8	742.0 ± 272.0 (*n* = 59)	−476.2	(−546.81; −405.59)	0.000	6
Shunwu et al., 2009 [20]	265.8 ± 104.8	711.4 ± 157.3 (*n* = 30)	−445.6	(−503.37; −387.83)	0.000	6.1
Lau et al., 2013 [21]	265.8 ± 104.8	661.0 ± 561.3 (*n* = 49)	−395.2	(−552.90; −237.50)	<0.001	5.4
Gu et al., 2014 [22]	265.8 ± 104.8	576.3 ± 176.2 (*n* = 38)	−310.5	(−368.01; −252.99)	0.000	6.1
Yang et al., 2017 [23]	265.8 ± 104.8	538.6 ± 129.5 (*n* = 20)	−272.8	(−331.02; −214.58)	0.000	6.1
Cheng et al., 2013 [24]	265.8 ± 104.8	535.5 ± 324.0 (*n* = 25)	−269.7	(−397.37; −142.03)	<0.001	5.7
Tschugg et al., 2017 [25]	265.8 ± 104.8	472.3 ± 555.0 (*n* = 48)	−206.5	(−364.04; −48.96)	0.01	5.4
Seng et al., 2013 [26]	265.8 ± 104.8	405.0 ± 80.0 (*n* = 40)	−139.2	(−167.19; −111.21)	0.000	6.2
Singh et al. 2014 [27]	265.8 ± 104.8	380.3 ± 191.2 (*n* = 33)	−114.5	(−181.02; −47.98)	<0.001	6.1
Villavicen et al., 2010 [28]	265.8 ± 104.8	366.8 ± 298.2 (*n* = 63)	−101	(−175.77; −26.23)	0.008	6
Adogwa et al., 2012 [29]	265.8 ± 104.8	280.0 ± 219.7 (*n* = 7)	−14.2	(−177.43; 149.03)	0.865	5.4
Pelton et al., 2012 [30]	265.8 ± 104.8	271.0 ± 84.9 (*n* = 33)	−5.2	(−36.95; 26.55)	0.748	6.2
Overall results			−331.23	(−432.01; −230.45)	<0.001 *	−6.442 *
Heterogeneity: I^2^ = 97.07%						
Homogeneity: Q = 714.992 *p*-value = 0.000						

* *p* and Z value overall effect size.

**Table 4 jcm-12-04755-t004:** Effect Size Estimates for Individual Studies (LOS (days)).

Study	MIS-TLIF Mean ± SD (*n* = 250)	OTLIF Mean ± SD (*n*)	Effect Size	Mean Diff, 95% CI	Sig. (2-Tailed)	Weight (%)
Lee et al., 2012 [15]	4.84 ± 0.81	6.80 ± 3.40 (*n* = 72)	−1.956	(−2.75; −1.16)	<0.001	7.2
Wang et al., 2010 [17]	4.84 ± 0.81	14.60 ± 3.80 (*n* = 43)	−9.756	(−10.9; −8.62)	0.000	7.1
Sulaiman et al., 2014 [18]	4.84 ± 0.81	0.20 ± 0.20 (*n* = 11)	4.644	(4.49; 4.8)	0.000	7.2
Zhang et al., 2017 [19]	4.84 ± 0.81	10.10 ± 3.20 (*n* = 59)	−5.256	(−6.08; −4.43)	0.000	7.2
Shunwu et al., 2009 [20]	4.84 ± 0.81	12.50 ± 1.80 (*n* = 30)	−7.656	(−8.31; −7.00)	0.000	7.2
Lau et al., 2013 [21]	4.84 ± 0.81	4.70 ± 2.10 (*n* = 49)	0.144	(−0.45; 0.74)	0.636	7.2
Gu et al., 2014 [22]	4.84 ± 0.81	12.10 ± 3.60 (*n* = 38)	−7.256	(−8.41; −6.11)	0.000	7.1
Cheng et al., 2013 [24]	4.84 ± 0.81	6.05 ± 1.80 (*n* = 25)	−1.206	(−1.92; −0.49)	<0.001	7.2
Tschugg et al., 2017 [25]	4.84 ± 0.81	19.10 ± 12.00 (*n* = 48)	−14.256	(−17.65; −10.86)	<0.001	6.5
Seng et al., 2013 [26]	4.84 ± 0.81	5.90 ± 0.40 (*n* = 40)	−1.056	(−1.22; −0.9)	0.000	7.2
Singh et al., 2014 [27]	4.84 ± 0.81	2.90 ± 1.10 (*n* = 33)	1.944	(1.56; 2.33)	0.000	7.2
Villavicen et al., 2010 [28]	4.84 ± 0.81	4.20 ± 3.50 (*n* = 63)	0.644	(−0.23; 1.51)	0.147	7.2
Pelton et al., 2012 [30]	4.84 ± 0.81	3v00 ± 1.10 (*n* = 33)	1.844	(1.46; 2.23)	0000	7.2
Kulkarni et al., 2016 [31]	4.84 ± 0.81	5.84 ± 2.25 (*n* = 25)	−0.996	(−1.88; −0.11)	0.028	7.2
Overall results			−2.78	(−5.45; −0.1)	0.042 *	−2.036 *
Heterogeneity: I^2^ = 99.08%						
Homogeneity: Q = 4420.336 *p* = 0.000						

* *p* and Z value for overall effect size.

**Table 5 jcm-12-04755-t005:** Effect Size Estimates for Individual Studies (Operative time (min)).

Study	MIS-TLIF Mean ± SD (*n* = 250)	OTLIF Mean ± SD (*n*)	Effect Size	Mean Diff, 95% CI	Sig. (2-Tailed)	Weight (%)
Lee et al., 2012 [15]	107.68 ± 29.23	181.80 ± 45.40 (*n* = 72)	−74.12	(−82.9; −65.34)	0.000	6
Wang et al., 2012 [16]	107.68 ± 29.23	168.00 ± 37.00 (*n* = 39)	−60.32	(−70.57; −50.07)	0.000	5.9
Wang et al., 2010 [17]	107.68 ± 29.23	145.00 ± 27.00 (*n* = 43)	−37.32	(−46.68; −27.96)	<0.001	6
Wang et al., 2010 [17]	107.68 ± 29.23	143.00 ± 35.00 (*n* = 27)	−35.32	(−47.16; −23.48)	<0.001	5.9
Sulaiman et al., 2014 [18]	107.68 ± 29.23	161.00 ± 7.60 (*n* = 11)	−53.32	(−70.65; −35.99)	<0.001	5.7
Zhang et al., 2017 [19]	107.68 ± 29.23	136.00 ± 25.00 (*n* = 59)	−28.32	(−36.4; −20.24)	<0.001	6
Shunwu et al., 2009 [20]	107.68 ± 29.23	142.80 ± 22.50 (*n* = 30)	−35.12	(−45.95; −24.29)	<0.001	5.9
Gu et al., 2014 [22]	107.68 ± 29.23	186.60 ± 23.40 (*n* = 38)	−78.92	(−88.66; −69.18)	0.000	5.9
Yang et al., 2017 [23]	107.68 ± 29.23	141.80 ± 18.80 (*n* = 20)	−34.12	(−47.16; −21.09)	<0.001	5.9
Cheng et al., 2013 [24]	107.68 ± 29.23	278.80 ± 14.50 (*n* = 25)	−171.12	(−182.73; −159.51)	0.000	5.9
Tschugg et al., 2017 [25]	107.68 ± 29.23	208.80 ± 86.00 (*n* = 48)	−101.12	(−114.56; −87.68)	0.000	5.8
Seng et al., 2013 [26]	107.68 ± 29.23	166.00 ± 7.00 (*n* = 40)	−58.32	(−67.43; −49.21)	0.000	6
Singh et al., 2014 [27]	107.68 ± 29.23	186.00 ± 31.00 (*n* = 33)	−78.32	(−89.01; −67.63)	0.000	5.9
Villavicen et al., 2010 [28]	107.68 ± 29.23	214.90 ± 60.00 (*n* = 63)	−107.22	(−117.57; −96.88)	0.000	5.9
Adogwa et al., 2012 [29]	107.68 ± 29.23	211.00 ± 43.23 (*n* = 7)	−103.32	(−125.58; −81.06)	0.000	5.5
Pelton et al., 2012 [30]	107.68 ± 29.23	184.50 ± 33.94 (*n* = 33)	−76.82	(−87.64; −66)	0.000	5.9
Kulkarni et al., 2016 [31]	107.68 ± 29.23	177.60 ± 34.20 (*n* = 25)	−69.92	(−82.13; −57.71)	0.000	5.9
Overall results			−70.27	(−87.61; −52.93)	<0.001 *	−7.943 *
Heterogeneity: I^2^ = 97.6%						
Homogeneity: Q = 673.156 *p* = 0.000						

* *p* and Z value for overall effect size.

**Table 6 jcm-12-04755-t006:** Effect Size Estimates for Individual Studies (ODI Change).

Study	MIS-TLIF Mean ± SD (*n* = 250)	OTLIF Mean ± SD (*n*)	Effect Size	Mean Diff, 95% CI	Sig. (2-Tailed)	Weight (%)
Lee et al., 2012 [15]	33.23 ± 23.94	23.70 ± 17.38 (*n* = 72)	9.528	(4.54; 14.52)	<0.001	7
Wang et al., 2010 [17]	33.23 ± 23.94	26.30 ± 6.42 (*n* = 43)	6.928	(3.39; 10.46)	<0.001	7.5
Wang et al., 2010 [17]	33.23 ± 23.94	26.40 ± 6.95 (*n* = 27)	6.828	(2.87; 10.79)	<0.001	7.4
Shunwu et al., 2009 [20]	33.23 ± 23.94	24.80 ± 10.57 (*n* = 30)	8.428	(3.62; 13.24)	<0.001	7.1
Gu et al., 2014 [22]	33.23 ± 23.94	28.40 ± 4.72 (*n* = 38)	4.828	(1.5; 8.16)	0.004	7.6
Yang et al., 2017 [23]	33.23 ± 23.94	30.70 ± 11.32 (*n* = 20)	2.528	(−3.25; 8.31)	0.391	6.7
Seng et al., 2013 [26]	33.23 ± 23.94	29.80 ± 11.85 (*n* = 40)	3.428	(−1.29; 8.15)	0.155	7.1
Brodano et al., 2015 [32]	33.23 ± 23.94	34.00 ± 7.07 (*n* = 34)	−0.772	(−4.57; 3.03)	0.691	7.4
Rodriguez-Vela et al., 2013 [33]	33.23 ± 23.94	9.09 ± 10.51 (*n* = 20)	24.138	(18.66; 29.62)	0.000	6.9
Adogwa et al., 2015 [34]	33.23 ± 23.94	14.88 ± 19.01 (*n* = 108)	18.348	(13.7; 23)	<0.001	7.1
Parker et al., 2012 [35]	33.23 ± 23.94	17.20 ± 1.58 (*n* = 15)	16.028	(9.91; 22.15)	<0.001	6.6
Adogwa et al., 2010 [36]	33.23 ± 23.94	17.20 ± 10.58 (*n* = 15)	16.028	(9.91; 22.15)	<0.001	6.6
Parker et al., 2013 [37]	33.23 ± 23.94	18.70 ± 9.39 (*n* = 50)	14.528	(10.58; 18.48)	<0.001	7.4
Peng et al., 2009 [38]	33.23 ± 23.94	30.20 ± 4.45 (*n* = 29)	3.028	(−0.35; 6.41)	0.079	7,5
Overall results			9.4	(5.68; 13.12)	<0.001 *	4.950 *
Heterogeneity: I^2^ = 99.08%						
Homogeneity: Q=116.364 *p* = 0.000						

* *p* and Z value for overall effect size.

## Data Availability

Data are available upon request from the corresponding author.

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
