# Peer review of "Latest Developments in Minimally Invasive Spinal Treatment in Slovakia and Its Comparison with an Open Approach for the Treatment of Lumbar Degenerative Diseases"

_jcm, 2023, doi:10.3390/jcm12144755_

Round 1
Reviewer 1 Report
The paper showed differences between two surgical procedures and how they can impact the quality of patient's life. Although they could not compare the differences directly by themselves, however the represented information can help us to understand the merit and demerit of the two approaches.
Here are some points that should be taken care.
1. Every abbreviation has to be written in elaborated form at their first using time. In abstract, every abbreviation was used without mentioning the full form of it. In the first paragraph of introduction, OTLIF is written without the full form. Same goes for the LOS mentioned in the aim section of the introduction.
2. The whole paper should be checked by a native English editor for proper use of punctuation marks, proper sentence making, and word choice as well. It will increase the level of understanding of the paper.
3. The first paragraph of introduction (The study is dealing ........ approach (OTLIF)) should be placed either in the later section of introduction or in the first paragraph of discussion. The second paragraph should be the starting point in the introduction.
4. Line 55-59. The sentence is too large to follow. Breaking down the sentence would help to understand and follow the meaning properly.
5. It is better to give more information about the background regarding this study instead of vast information of postoperative rehabilitation. Introduction is not discussion. Introduction should precisely and briefly consist of the background, the problem, hint of a solution and aim of the study. Not the information about health insurance providing spa facilities.
6. Use appropriate headings to related paragraphs in discussion.
7. Heading of the tables should follow one style. In table 1 only first word started with capital letter, whereas in table 2 every word started with capital letter.
8. Use a consistent style to present p value. In someplace, p value is showed as .00 an in someplace as ,00
9. Line 404-408. Breakdown the sentence for clarification of the statement.
10. Line 456. is the appropriate word founding or finding?
Please send this paper to a good English editing company so that the given information can easily be understood.
Author Response
Dear reviewers and editor,
We sincerely thank you for the comments and suggestions. We incorporated them into our article. You will find them highlighted in the text. We must state, that the adding of your comments into the article raised its quality to a higher level. At the same time, we hope that the modification will meet Your expectations.
In the next section, we describe the adjustment of changes in detail.
The entire text was checked by our colleague Associate Professor doc. Andrey Kraev, Csc., who is the head of the Department of Foreign Languages at our University and specializes in the English language (https://www.ku.sk/en/faculties/faculty-of-education/departments/department-of-foreign-languages/employees.html)
Changes for Reviewer 1 are marked with green colour in the text.
Changes for Reviewer 2 are marked with blue colour in the text.
Changes for Reviewer 3 are marked with grey colour in the text.
Reviewer 1
Comments and Suggestions for Authors
The paper showed differences between two surgical procedures and how they can impact the quality of patient's life. Although they could not compare the differences directly by themselves, however the represented information can help us to understand the merit and demerit of the two approaches.
Here are some points that should be taken care.
- Every abbreviation has to be written in elaborated form at their first using time. In abstract, every abbreviation was used without mentioning the full form of it. In the first paragraph of introduction, OTLIF is written without the full form. Same goes for the LOS mentioned in the aim section of the introduction.
Response: All abbreviations used in abstract are given in full form when first used.
- The whole paper should be checked by a native English editor for proper use of punctuation marks, proper sentence making, and word choice as well. It will increase the level of understanding of the paper.
Response:
- The first paragraph of introduction (The study is dealing ........ approach (OTLIF)) should be placed either in the later section of introduction or in the first paragraph of discussion. The second paragraph should be the starting point in the introduction.
Response: The first paragraph of the introduction is deleted.
- Line 55-59. The sentence is too large to follow. Breaking down the sentence would help to understand and follow the meaning properly.
Response: The sentence is break down.
- It is better to give more information about the background regarding this study instead of vast information of postoperative rehabilitation. Introduction is not discussion. Introduction should precisely and briefly consist of the background, the problem, hint of a solution and aim of the study. Not the information about health insurance providing spa facilities.
Response: We excluded general information from the introduction. We have supplemented the introduction with specific of minimally invasive technique and we described the implementation of MIS-TLIF at our clinic. On the recommendation of one of the reviewers, we add the description of the rehabilitation in Methods.
- Use appropriate headings to related paragraphs in discussion.
Response: Headings are assigned to paragraphs.
- Heading of the tables should follow one style. In table 1 only first word started with capital letter, whereas in table 2 every word started with capital letter.
Response: Heading of the tables follow one style. Table 1 and 2 were correct.
- Use a consistent style to present p value. In someplace, p value is showed as .00 an in someplace as ,00
Response: We consistent style to present p value. We use = .00.
- Line 404-408. Breakdown the sentence for clarification of the statement.
Response: The sentence is break down.10.
Line 456. is the appropriate word founding or finding?
Response: appropriate word is finding.
Comments on the Quality of English Language
Please send this paper to a good English editing company so that the given information can easily be understood.
Response: The entire text was checked by our colleague Associate Professor doc. Andrey Kraev, Csc., who is the head of the Department of Languages at our University and specializes in the English language.
Reviewer 2 Report
Potasova and colleagues report a series of 250 minimally invasive spinal interbody fusions performed at their center and compare their results to published data for open lumber interbody fusions. Their scientific ambition of comparing their technique and performance to the current international standard of care (open) is commendable. While the manuscript in principle elucidates interesting findings, major revisions are warranted before considering publication.
Methodological remarks:
The authors select a number of OTLIF studies to compare their minimally invasive approach against. It would be interesting to see, how their results measure against the published data regarding minimally invasive techniques.
Introduction:
The introduction is lengthy and at times not really on point. I would recommend cutting significant portions of general introduction about back pain and degenerative spinal disorders and rather elaborate on the specific minimally invasive technique Potsova and colleagues apply at their center.
Methods:
Inclusion and Exclusion criteria should be presented in tabular form, this would make it easier to read than the prosaic style.
Figure 1 is superfluous, as all information is already given in the text.
I suggest one dataset of studies for all parameters, rather than choosing 17 articles for the first set of parameters and 14 for the second set.
Results:
In Table 1, only „most frequenlty operated segment“ is mentioned, this is problematic for two reasons, (1) l4-S1 is not one segment (2) information should be provided on frequency of each instrumented segment, i.e. L1/2: X operations, etc.
Tables 2-5 are supposed to show the data from the meta-analysis. However, the absolute values are missing and only effect sizes reported. I suggest to add the absolute values extracted from the studies. Further, the authors should consider representing the data in Tables 2-5 in a graphical manner for better comprehension.
Discussion:
The discussion appears to largely reiterate findings already presented in the results, both in the text as well as in the tables. This should be extensively reworked and a special focus put on comparing the results of this current study with other studies comparing minimally invasive techniques with open surgical techniques.
Limitations should be separate section.
I recommend extensive language editing by a native speaker or professional editing service, especially the discussion is very difficult to understand.
Author Response
Dear reviewers and editor,
We sincerely thank you for the comments and suggestions. We incorporated them into our article. You will find them highlighted in the text. We must state, that the adding of your comments into the article raised its quality to a higher level. At the same time, we hope that the modification will meet Your expectations.
In the next section, we describe the adjustment of changes in detail.
The entire text was checked by our colleague Associate Professor doc. Andrey Kraev, Csc., who is the head of the Department of Foreign Languages at our University and specializes in the English language (https://www.ku.sk/en/faculties/faculty-of-education/departments/department-of-foreign-languages/employees.html)
Changes for Reviewer 1 are marked with green colour in the text.
Changes for Reviewer 2 are marked with blue colour in the text.
Changes for Reviewer 3 are marked with grey colour in the text.
Reviewer 2
Comments and Suggestions for Authors
Potasova and colleagues report a series of 250 minimally invasive spinal interbody fusions performed at their center and compare their results to published data for open lumber interbody fusions. Their scientific ambition of comparing their technique and performance to the current international standard of care (open) is commendable. While the manuscript in principle elucidates interesting findings, major revisions are warranted before considering publication.
Methodological remarks:
The authors select a number of OTLIF studies to compare their minimally invasive approach against. It would be interesting to see, how their results measure against the published data regarding minimally invasive techniques.
Introduction:
1.The introduction is lengthy and at times not really on point. I would recommend cutting significant portions of general introduction about back pain and degenerative spinal disorders and rather elaborate on the specific minimally invasive technique Potsova and colleagues apply at their center.
Response: We excluded general information from the introduction. We have supplemented the introduction with specific of minimally invasive technique and we described the implementation of MIS-TLIF at our clinic. On the recommendation of one of the reviewers, we put the description of the rehabilitation in Methods.
Methods:
2.Inclusion and Exclusion criteria should be presented in tabular form, this would make it easier to read than the prosaic style.
Response: We present Inclusion and Exclusion criteria in tabular form (Table 1).
3.Figure 1 is superfluous, as all information is already given in the text.
Response: Figure 1 is deleted.
4.I suggest one dataset of studies for all parameters, rather than choosing 17 articles for the first set of parameters and 14 for the second set.
Response: We give one dataset of studies for all parameters.
Results:
5.In Table 1, only „most frequenlty operated segment “is mentioned, this is problematic for two reasons, (1) l4-S1 is not one segment (2) information should be provided on frequency of each instrumented segment, i.e. L1/2: X operations, etc.
Response: The Table is modified. It lists all operated levels of lumbar spine performed in our clinic in 2022. We present results as number of patients and percentages for each treated segment / segments. As we have added a table for Inclusion and Exclusion criteria, results for treated segments can be found in Table 2.
6.Tables 2-5 are supposed to show the data from the meta-analysis. However, the absolute values are missing and only effect sizes reported. I suggest to add the absolute values extracted from the studies. Further, the authors should consider representing the data in Tables 2-5 in a graphical manner for better comprehension.
Response: Tables 2 - 5 are modified. As we have added Table 2, the changes can be found in Tables 3-6). Absolute values are supplemented. For the clarity of the results, we include graphs in the study as well. This graphical representation of the results can serve for better comprehension of the results, as the reviewer suggested.
Discussion:
7.The discussion appears to largely reiterate findings already presented in the results, both in the text as well as in the tables. This should be extensively reworked and a special focus put on comparing the results of this current study with other studies comparing minimally invasive techniques with open surgical techniques.
Limitations should be separate section.
Response: In the discussion, we compare our results with other 10 studies, as the reviewer requested. Limitations are on separate section
8.Comments on the Quality of English Language
I recommend extensive language editing by a native speaker or professional editing service, especially the discussion is very difficult to understand.
Response: The entire text was checked by our colleague Associate Professor doc. Andrey Kraev, Csc., who is the head of the Department of Languages at our University and specializes in the English language.
Reviewer 3 Report
The authors investigated blood loss, operation time, LOS and ODI index for patients with degenerative lumbar diseases after MIS-TLIF. Then, they compared these results of MIS-TILF group with that of OTLIF group. They used control data of OTLIF from literature because they have no longer carried out OTLIF for ten years. They mentioned the benefit of MIS-TLIF, such as lower blood loss, shorter operation time, shorter LOS and greater change of ODI. These results must be beneficial for the patients with lumbar degenerative diseases. However, I have following concerns.
Major issues
(1) I understood that the authors showed MIS-TLIF group presented lower blood loss, shorter operation time, shorter LOS and greater change of ODI than those of OTLIF group. There are several reports of comparison between MIS-TLIF and OTLIF (Hu X, et al. Global Spine J. 2023, Heemskerk JL, et al. Spine J 2021 21:2049-2065 and Hammad A, et al. J Orhop 2019, https://doi.org/10.1186/s13018-019-1266-y). Their results varied on the point of operation time, VAS, JOA score, ODI and long term outcome. In this article, the authors often tried to explain with high degree of heterogeneity. I would like the authors to discuss based on these previous works.
(2) I would like the authors to emphasize something new in this article. Many reports mentioned minimal invasive spine surgery obtained the results of lower blood loss, shorter LOS and greater ODI changes.
(3) The authors have presented better changes of ODI in MIS-TLIF group. They expressly mentioned importance of complex postoperative rehabilitations after MIS-TLIF. They had better focus on content of post-operative rehabilitations in the control groups from literature.
Minor concerns
(1)I felt ‘introduction’ part was long-winded. Contents of operative procedure and rehabilitation program (from line 68 to 96) should be transferred into ‘Material and Methods’ parts.
(2) What is LOS? LOS means ‘length of hospital stay’? The authors should mention abbreviation.
(3) In Table 1, the authors should describe not only L4/5/L1 but also all other operation lesions of 159 patients.
(4) In Discussion part, MISS-TLIF should be corrected into MIS-TLIF.
(5)In ‘Reference’ part, the authors must write official journal abbreviations. For example, ‘Current reviews in musculoskeletal medicine’ should be changed into ‘Curr. Rev. Musculoskelet’. Please check other journals in Reference part.
Author Response
Dear reviewers and editor,
We sincerely thank you for the comments and suggestions. We incorporated them into our article. You will find them highlighted in the text. We must state, that the adding of your comments into the article raised its quality to a higher level. At the same time, we hope that the modification will meet Your expectations.
In the next section, we describe the adjustment of changes in detail.
The entire text was checked by our colleague Associate Professor doc. Andrey Kraev, Csc., who is the head of the Department of Foreign Languages at our University and specializes in the English language (https://www.ku.sk/en/faculties/faculty-of-education/departments/department-of-foreign-languages/employees.html)
Changes for Reviewer 1 are marked with green colour in the text.
Changes for Reviewer 2 are marked with blue colour in the text.
Changes for Reviewer 3 are marked with grey colour in the text.
Reviewer 3
Comments and Suggestions for Authors
The authors investigated blood loss, operation time, LOS and ODI index for patients with degenerative lumbar diseases after MIS-TLIF. Then, they compared these results of MIS-TILF group with that of OTLIF group. They used control data of OTLIF from literature because they have no longer carried out OTLIF for ten years. They mentioned the benefit of MIS-TLIF, such as lower blood loss, shorter operation time, shorter LOS and greater change of ODI. These results must be beneficial for the patients with lumbar degenerative diseases. However, I have following concerns.
Major issues
- I understood that the authors showed MIS-TLIF group presented lower blood loss, shorter operation time, shorter LOS and greater change of ODI than those of OTLIF group. There are several reports of comparison between MIS-TLIF and OTLIF (Hu X, et al. Global Spine J. 2023, Heemskerk JL, et al. Spine J 2021 21:2049-2065 and Hammad A, et al. J Orhop 2019, https://doi.org/10.1186/s13018-019-1266-y). Their results varied on the point of operation time, VAS, JOA score, ODI and long term outcome. In this article, the authors often tried to explain with high degree of heterogeneity. I would like the authors to discuss based on these previous works.
Response: We edited the discussion. We also included other studies and discussed the results with these studies.
- I would like the authors to emphasize something new in this article. Many reports mentioned minimal invasive spine surgery obtained the results of lower blood loss, shorter LOS and greater ODI changes.
Response: This is added to section: “A strength of the study “
- The authors have presented better changes of ODI in MIS-TLIF group. They expressly mentioned importance of complex postoperative rehabilitations after MIS-TLIF. They had better focus on content of post-operative rehabilitations in the control groups from literature.
Response: This is added in discussion in part
Minor concerns
- I felt ‘introduction’ part was long-winded. Contents of operative procedure and rehabilitation program (from line 68 to 96) should be transferred into ‘Material and Methods’ parts.
Response: We add the description of the rehabilitation in Methods. We excluded general information from the introduction. We have supplemented the introduction with specific of minimally invasive technique and we described the implementation of MIS-TLIF at our clinic.
(2) What is LOS? LOS means ‘length of hospital stay’? The authors should mention abbreviation.
Response: All abbreviations used in abstract are given in full form when first used. Yes, LOS means ‘length of hospital stay’
(3) In Table 1, the authors should describe not only L4/5/L1 but also all other operation lesions of 159 patients.
Response: The Table is modified. It lists all operated levels of lumbar spine performed in our clinic in 2022. We present results as number of patients and percentages for each treated segment / segments. As we have added a table for Inclusion and Exclusion criteria, as one reviver recommend, results for treated segments can be found in Table 2.
- In Discussion part, MISS-TLIF should be corrected into MIS-TLIF.
Response: It is corrected.
- In ‘Reference’ part, the authors must write official journal abbreviations. For example, ‘Current reviews in musculoskeletal medicine’ should be changed into ‘Curr. Rev. Musculoskelet’. Please check other journals in Reference part.
Response: It is corrected.
Round 2
Reviewer 2 Report
The authors have done an admirable job revising the manuscript extensively and have addressed all points. I recommend publication.
One typo; Fig.1 "Exclused" should be "excluded"
Reviewer 3 Report
The authors revised manuscript properly. I would like the editor to accept the article for the journal.